# The Histopathological Characteristic of Gastric Carcinoma in the Belgian Tervueren and Groenendael Dog: A Comparison of Two Classification Methods

**DOI:** 10.3390/ani13091532

**Published:** 2023-05-03

**Authors:** Christina Kijan, Sanne Hugen, Rachel E. Thomas, Anita M. Oberbauer, Peter A. J. Leegwater, Hille Fieten, Alexander J. German, Paul J. J. Mandigers

**Affiliations:** 1Expertise Centre Genetics, Department of Clinical Sciences, Faculty of Veterinary Medicine, Utrecht University, 3584 CM Utrecht, The Netherlands; c.kandanearachigedon@uu.nl (C.K.);; 2Department of Pathobiology, Faculty of Veterinary Medicine, Utrecht University, 3584 CM Utrecht, The Netherlands; 3Department of Animal Science, University of California, Davis, CA 95616, USA; 4Institute of Life Course and Medical Sciences, University of Liverpool, Leahurst Campus, Wirral, Neston CH64 7TE, UK; 5IVC Evidensia Referral Hospital Arnhem, Meander 10, 6825 MB Arnhem, The Netherlands

**Keywords:** stomach, carcinoma, canine, pathology, Belgian Shepherd dog, Tervueren, Groenendael

## Abstract

**Simple Summary:**

Gastric carcinoma is a type of stomach cancer that occurs more frequently in Tervueren and Groenendael dogs compared to other breeds. In this study, gastric tumour biopsies of 61 Belgian Shepherd dogs were examined for inflammation and tumour location, and subsequently classified using the World Health Organization (WHO) and Laurén classifications, which are commonly used in classifying human gastric carcinoma, as there is no gold standard for the histological evaluation of canine gastric carcinoma as of yet. Survival time after the onset of symptoms was recorded and was investigated in relation to tumour classification. An intestinal type (according to the Laurén classification) and a tubular tumour pattern (according to the WHO classification) were both associated with a longer median survival time. This may aid the practising veterinarian as a prognostic tool for canine gastric carcinoma and provide information regarding the use and reliability of the use of both scoring systems.

**Abstract:**

Gastric carcinoma is generally considered to be a rare disease in dogs, carrying a grave prognosis. However, in the Tervueren and Groenendael varieties of the Belgian Shepherd dog breed, the disease is highly prevalent. While histopathology is the gold standard for diagnosing gastric carcinoma, there is no general consensus on the methods for histological classification in these cases. Biopsies of a group of 61 dogs with confirmed gastric carcinoma (45 Tervueren and 16 Groenendael) were examined and classified according to World Health Organization (WHO) and Laurén classifications. Kaplan–Meier curves were used to compare survival between the different subtypes and simple and multiple linear regression were used to analyse the association between age of onset and breed variant, sex, neuter status, location of the tumour, inflammation score, and Laurén and WHO classifications. Mean age at diagnosis was significantly different in Groenendael (10.1 ± 2.01) and Tervueren dogs (8.5 ± 1.90). The Laurén classification resulted in 29 (48%) diffuse- and 32 (52%) intestinal-type tumours. Applying the WHO classification resulted in 30 (49%) tubular carcinoma growth patterns and 31 (51%) others. Median survival time was significantly reduced for the diffuse type as compared to the intestinal type according to the Laurén classification, with the same median survival time results for tubular compared to non-tubular subtypes according to the WHO classification (median survival time of 61 vs. 182 days, respectively). Using the WHO and Lauren classification on tumour biopsies may help the practising clinician in the prognostication of gastric carcinoma in Tervueren and Groenendael dogs.

## 1. Introduction

Gastric cancers (including adenocarcinoma, leiomyosarcoma, gastrointestinal stromal tumour and lymphoma) account for <1% of all tumours in dogs and are typically malignant. Gastric adenocarcinoma (GC) is the most prevalent type of gastric cancers. The prognosis in dogs with gastric neoplasia is poor, with median survival times reported of just 33 to 72 days after starting treatment [1,2,3]. Metastasis at the time of diagnosis is reported to be 70–90% [1,4,5], and the most common sites of metastasis are the regional lymph nodes, omentum, duodenum, liver, pancreas, spleen, oesophagus, adrenal glands and lungs [1,4,5,6]. Surgical management is often precluded by the tumour location at the curvature minor of the stomach and, even when performed at early stages of the disease, the response is usually poor, and adjuvant chemotherapy has not been shown to provide additional or curative value [1,7,8].

The strong breed predisposition of GC in Belgian Shepherds is indicative of a hereditary component; however, the genetic background has not yet been elucidated [9]. In Tervueren and Groenendael dogs, a high incidence and a likely genetic predisposition has been previously described [4,10,11,12,13,14]. It is, however, unknown if these dogs truly belong to one phenotype, or if different subtypes of gastric carcinoma can be relevant for prognosis and possible aetiology.

Gastroscopy with the collection of biopsies for histopathological examination is the diagnostic standard in dogs suspected of gastric tumours. The classification of histological biopsies can be an important tool for the prognostication of tumours; however, a general consensus regarding the histological evaluation of canine GC has not been described in the literature. Various tumour classification methods exist for human gastric carcinoma, including the WHO classification and the Laurén classification, with the WHO classification being the most widely used method [15,16,17]. The Laurén classification system and an amended scheme based on WHO classifications are also mostly used for the histopathology of GC in domestic animals [18]. The Laurén classification has been applied to gastric carcinoma in dogs in past studies too, providing a basis for a past WHO classification in 1976 [19]. Research conducted in 1978 with a sample size of 26 found that the two main histologic types of human gastric adenocarcinoma also occurred in dogs [20]. In this study, the following WHO categories according to growth pattern [15,16,18] were used: tubular carcinoma, mucinous carcinoma, signet ring cell carcinoma, papillary carcinoma and undifferentiated carcinoma. Alternative to using the WHO classification, staging in human gastric carcinoma cases can follow the Laurén classification into intestinal type or diffuse type. Differentiation between the types is based on general structure, cell structure, secretion, growth pattern and mode of growth [17]. Even though both the Laurén and WHO staging systems are used in animals as well, the clinical significance of the classification of gastric carcinoma in dogs is currently unclear [18,21].

While the role of inflammatory cells has not yet been fully studied in canine gastric carcinomas, and therefore the potential of inflammation score as a prognostic factor is not yet known, the investigation of inflammatory infiltrate may be of importance, as inflammatory cells play a complex and multifaceted role in the aetiology of gastric carcinoma in humans [22]. In humans, *Helicobacter pylori* infections are an additionally known risk factor for the development of GC that can be confirmed by histopathology [23,24,25]. *H. pylori* causes both cell proliferation and gastric inflammation, predisposing the gastric lining to neoplasia. However, there has not been any evidence found of *H. pylori* playing a role in the development of GC in canines [9,26,27,28]. In addition, while GC mostly occur in the curvature minor, they can expand to or start in other anatomical regions of the stomach as well, such as the cardia. It is currently unknown if tumours occurring at a region different from the minor curvature constitute the same or a different clinicopathological group.

The main aim of the current study is the comparison between the WHO and the Laurén classification schemes and whether they are of influence on the age of onset and survival in 61 cases of gastric carcinoma in the Tervueren and Groenendael variants of the Belgian Shepherd dog breed. Additionally, we investigated the influence of the involvement of the curvature minor or cardia, inflammation status and the presence of Helicobacter-like organisms, thereby exploring the function of tumour classifications and other pathological factors as prognostic indicators. This information will shed new light on the added value of histological biopsies for veterinary clinicians with cases of canine gastric carcinoma, thus aiding the practicing clinician in optimizing individual treatment plans, substantiating the election for further surgical excision or euthanasia, and will contribute to a growing consensus on the histopathological classification of canine gastric carcinoma.

## 2. Materials and Methods

### 2.1. Patient Collection

Histopathological classification was performed on 61 cases of canine GC. Forty-three of the study dogs were identified from Belgian Shepherd dogs in the Netherlands, referred for gastroduodenoscopy to one of the authors (PJJM), between 2003 and May 2017, whilst the remaining 18 dogs were reported to the authors from owners, breeders and referring veterinarians. For the latter cases, veterinarians were asked to send biopsy material for assessment. Information regarding the patients (onset of symptoms, patient description) was obtained via questionnaire from the owners and through a form and patient files (endoscopic findings, diagnosis) from the practitioner, in the case of referrals. Thirteen different veterinarians participated in the referral of the 18 externally obtained cases.

### 2.2. Gastroscopy and Collection of Biopsies

Preparation for the gastroscopy (Figure 1) as well as the endoscopy itself was performed as described previously [29]. The complete stomach was examined macroscopically and if an abnormality was observed, 3–6 biopsies were obtained from both the edges and centre of the visually abnormal tissue. If a post-mortem examination was conducted, biopsies were taken during necropsy. The location of the tumour was noted per case, focussing on whether or not the curvature minor and cardia were involved, as these are specific locations that often heavily complicate surgical excision. Biopsies of normal tissue were not included in this study. In total, the samples of 17 out of 61 dogs were taken during necropsy. For these cases, this resulted in large biopsies.

### 2.3. Histological Evaluation

All biopsies were fixed in formalin and processed routinely, and 10 µm thick sections were stained according to standard procedures with haematoxylin and eosin. All slides from gastric carcinoma cases available were re-examined by a single pathologist (RT) who was not aware of the original histopathological diagnosis. Slides were intermixed with 15 gastritis cases, all Belgian Shepherd dogs, to ensure complete blinding.

### 2.4. Histopathology

All slides of gastric carcinoma biopsies were scored according to the WHO classification (Figure 2), according to the criteria in Table 1, and according to the Laurén classification as used for dogs into the intestinal subtype and the diffuse subtype. In cases where more than one WHO cell type was present, the predominant pattern or cell type was used to classify the tumours. Slides were also scored for the presence of gastric spiral-like organisms (=GHLO; gastric Helicobacter-like organisms) and, if present, whether superficial or intraglandular. The degree of inflammation associated with the gastric tumour was determined according to the reference values for normal canine superficial gastric leucocytes published by Day et al. [30], in which respective values of intraepithelial lymphocytes, lamina propria lymphocytes, lamina propria eosinophils and lamina propria plasma cells are listed. The data from this scheme were extrapolated to be used on the tumour samples. Inflammation status was scored as either normal (normal intraepithelial lymphocytes; sparse population of approximately 1–2 cells per stretch of 50 epithelial cells), mild (a mild increase in intraepithelial lymphocytes; individual lymphocytes, up to 10 per stretch of 50 epithelial cells), moderate (a moderate increase in intraepithelial lymphocytes; lymphocytes may cluster in groups of up to 4 cells), or severe (increase in intraepithelial lymphocytes; epithelium is more diffusely infiltrated by lymphocytes, the same applies to the neutrophilic and eosinophilic granulocyte component. There may be up to 20 per stretch of 50 epithelial cells) [30]. No exact cell counts were documented though, and the eventual scoring was subjectively based on the opinion of the pathologist during histology.

### 2.5. Statistical Analysis

Computer software was used for data analysis (R Version 4.2.2, R-studio, 31 October 2022), with the level of statistical significance set at *p* < 0.05 for 2-sided analyses. The significance of the veterinarians involved in the biopsies (cases from PJJM vs. those from referring veterinarians) was first tested via a simple linear regression with the clinics and age of onset (based on time of formal diagnosis by endoscopy) as explanatory variables. Either Wilcoxon rank sums tests or two-sample T-test were used to explore differences in onset for sex, variant and neuter status based on distribution of the data. Age of onset was reported as mean ± standard deviation. Simple linear regression was used to test a relation between age of onset (dependent variable) with sex, variant, neuter status, WHO subtype, Laurén classification, inflammation status and involvement of the lesser curvature or cardia as explanatory (independent) variables. Linear regression for WHO subtype was performed twice, either by grouping all non-tubular types (subtype 2) or by combining the tubulopapillary type with the tubular type (subtype 3), as the tubulopapillary growth pattern is a mixed type. The validity of models was tested by confirming that residuals were normally distributed (using QQ-plots and the Shapiro–Wilk test), as well as using the non-constant error variance test and Breusch–Pagan test (for homoscedasticity). Influential data points were examined using Cook’s distance. Associations between age and explanatory variables were further explored using multiple regression, by including different combinations of explanatory variables, with multi-collinearity tested using variance inflation factors (VIFs) and the best-fit model determined using the Bayesian information criterion (BIC). Significant variables were then tested using the chi-squared test for associations with either classification method.

Survival analysis was used to determine the survival time from the first signs of gastric cancer as described by the owner, with the primary endpoint being death from gastric cancer. Major outliers were identified and removed from the analysis. A stratified Cox’s proportional hazard model was used to stratify the data, with referring veterinarians as the stratification variable (cases from PJJM vs. those from referring veterinarians). This added a random effect in the models to correct for any biases related to having cases from different veterinarians. Initially, Kaplan–Meier curves were created to assess the association of explanatory variables (e.g., variant, sex, neuter status, WHO subtype, Laurén classification, inflammation status and involvement of the lesser curvature or cardia) and survival. Survival was further assessed with simple and multiple Cox’s regression analysis, stratified according to referring veterinarian as explained above. Given that none of the dogs were alive at time of analysis, and all dogs died with GC as the cause of death, there were no censored data. This includes owner-elected euthanasia due to poor prognosis and clinical presentation. Each variable was tested using simple regression, with variables of significance on simple Cox’s regression being entered into a multiple regression model. This model was further stratified using backwards elimination of variables, with multi-collinearity tested using VIFs and the best-fit model determined using the BIC. The assumptions of the Cox proportional hazards model were tested using Schoenfeld residuals. The level of statistical significance was set at *p* < 0.05 and tests were two-sided.

## 3. Results

### 3.1. Patient Data

Histopathological data were available from 61 Belgian Shepherd dogs, with 45 belonging to the Tervueren and 16 to the Groenendael variants, respectively. The dataset comprised 30 females (20 neutered) and 31 males (15 neutered, one unknown) (Table 2). Details regarding the age of onset of all dogs’ signs or symptoms were provided by the owners by questionnaire. These symptoms could consist of vomiting, anorexia, diarrhoea, polyuria/polydipsia and weight loss, and these data were further used as the onset within the survival analysis. No intent to curative treatment, other than purely palliative, was employed in any of the dogs included in this study. They were managed palliatively with stomach lining protectants, gastric acid inhibitors and anti-emetics. Further patient information and findings from endoscopic examination were provided by either question form or the patient file by the practitioner. Information regarding the location of the tumour was available for 52 dogs, whilst in 29 dogs, the degree of tumour-associated inflammation was scored. An outlier was identified and removed within the analysis. Simple linear regression analysis revealed a difference in age of onset at diagnosis between breed variants (*p* = 0.009), with Tervueren dogs (8.5 ± 1.90 years) being diagnosed at a younger age than Groenendael dogs (10.1 ± 2.01 years). The two different breed variants were not associated with either Laurén classification (*p* = 0.891) or WHO classification (*p* = 0.654). Mean age at diagnosis was 8.9 ± 2.01 years for all dogs (Figure 3), and the onset of symptoms was, on average, 110 days before diagnosis.

### 3.2. Tumour Classification and Pathology

All gastric tumour specimens were gastric carcinomas; no other tumour type was present in any of the gastric tumours in the Belgian Shepherds. All 61 samples were scored based on the WHO criteria (Table 1) and the Laurén classification, resulting in 29 (48%) dogs that were characterized as the diffuse type and 32 (52%) as the intestinal type. When classifying according to the WHO criteria, 12 (20%) were classified as mucinous, 5 (8%) as signet-ring type, 30 (49%) as tubular, 0 (0%) as papillary, 2 (3%) as tubulopapillary and 12 (20%) as unclassified. As not enough of the non-tubular types were available in the database, all non-tubular types were grouped together (subtype 2) or all types were grouped together excluding the tubular and tubulopapillary types (subtype 3). Further analysis regarding the WHO classification was completed by dividing the tubular type and subtype 2. As only breed variant was significant in the univariable linear regression analysis, no multiple regression analysis was performed for age of onset.

The curvature minor (CM) and cardia were involved in 42 (69%) and 4 (7%) dogs (Table 2), respectively, and there was no difference in the age of onset between these groups. There was no difference in either age at diagnosis (*p* = 0.400) or survival time (*p* = 0.998) between CM-positive (mean age, 9.0 years ± 2.14; survival time, 135 days (7–730 days)) and CM-negative (mean age, 8.4 years ± 1.90; survival time, 155 days (55–379 days)) dogs. Similarly, there was no association with either age at diagnosis (*p* = 0.780) or survival time (*p* = 0.663) between dogs with tumours involving (mean age, 8.6 years ± 1.47; survival time, 155 days (121–379 days)) or not involving (mean age, 8.9 years ± 2.18; survival time, 134 days (7–730 days)) the cardia. The results of the univariable linear regression analyses are summarised in Table 3. In some cases, the clinical records contained a note regarding ulceration of the associated gastric mucosa, and such changes could usually also be identified on histological examination. An association between both classification schemes was determined (*p* = 2793 × 10^−12^). Twenty-nine (48%) tumours were labelled as diffuse according to the Laurén classification, and these were all non-tubular according to WHO classifications. Thirty (49%) tumours were labelled as intestinal according to the Laurén classification, and these were all tubular according to WHO classifications. Two (3%) of the tumours labelled as diffuse through the Laurén classifications were scored as non-tubular with the WHO classifications; however, these were mixed types. Histological classification according to both the WHO and Laurén criteria is reported in Table 4. GHLO were reported in two of the biopsies (3%), with a superficial pattern reported in one case and an intraglandular pattern in the other. The tumour-associated inflammation score was mild (1) in 18 cases (30%), moderate (2) in 39 cases (64%), and severe (3) in 2 cases (3%). Two cases received no inflammation score due to the amount of necrosis present. Because of low numbers, the inflammation scores for moderate and severe were merged. No significant association was found between inflammation score and age of onset. No further analyses were conducted in regard to specific inflammatory cells. Because GHLO were only detected in 2 out of 61 dogs, statistical analysis was not possible for this variable.

### 3.3. Survival Analysis

One female dog from the 61 histologically scored dogs was identified as an outlier in the survival analysis, with a survival time of 1553 days after the first signs of GC appeared. Although the cause of death was proven to be due to GC, the initial gastrointestinal issues occurred 730 days before the formal diagnosis and, therefore, might have been a consequence of another underlying disease. Given that a lead time to diagnosis of over a year has only been observed in one other case in the database, this was assumed to be due to a data error and, as a result, the dog was removed from further analysis.

In the multivariable Cox proportional hazard model for total survival time after the start of symptoms with WHO and Laurén classification as variables (significant in the univariable Cox PH models), both did not have an effect (*p=* 0.379 and *p* = 0.822, respectively). This might be because these variables are correlated. In simple regression, based on the Laurén classification, the median total survival time was shorter for dogs with diffuse gastric carcinoma (median 61 days) compared to intestinal-type gastric carcinoma (median, 182 days; Table 5; Figure 4; *p*= 0.006). There were similar findings when survival time was assessed according to the WHO classifications for the tubular type compared with non-tubular types combined (*p=* 0.004) (Table 6; Figure 5). There was no significant influence of sex, neutering, tumour location, severity of inflammation or complete WHO staging on age of onset or survival. Total survival time was not significantly associated with the age of onset.

## 4. Discussion

The Belgian shepherd dog breed comprises four varieties: Malinois, Laekenois, Tervueren and Groenendael. Dogs of the Tervueren and Groenendael varieties are often crossed with one another, and recent work using principal component analysis has demonstrated complete overlapping of the Tervueren and Groenendael family clusters in [31,32]. The authors have observed that gastric carcinomas are highly prevalent in both Tervueren and Groenendael shepherd dogs, whilst there can be both early and late onset within families with a high prevalence of the disease. In the current study, we characterised gastric carcinoma in a group of 61 confirmed cases of Belgian shepherd dogs. The relatively high prevalence and poor prognosis within this breed suggests that exploring a genetic basis for the condition can provide insight into reducing the disease. Mean age of onset was similar to findings in previously reported case series of gastric carcinoma in Belgian shepherd dogs [4] and other breeds [21,33]. In contrast to these earlier, smaller studies, we did not find a male predisposition, as previously reported [9].

Mean age at diagnosis is 8.9 years in these breed varieties (Figure 3). A significant difference in the age of onset for the Groenendael and Tervueren varieties in the Belgian shepherd dogs was found, with Tervueren dogs being diagnosed at a younger age. However, this finding should further be investigated in a cohort with more Groenendael dogs and a more equal distribution between the varieties. As no association was found between the breed variants and the classification schemes, a cause for this difference in age could be variant specific.

The gold standard of diagnosis is biopsies taken during gastroscopy, laparotomy or during post-mortem examination. The lesion typically involves the curvature minor of the stomach (69% of cases) and this location, in combination with the advanced stage of the disease at the time of diagnosis, renders curative surgery almost always impossible. When more than one cell type was present in a case, the predominant pattern or cell type was used for classification. As total representations of the tumours were usually not available, a compromise was made by ensuring that every case had at least three different biopsies to heighten the predictive value of the biopsies for the rest of the mass. Prognosis is poor with a mean survival from time of first symptoms to death of approximately 6 months (182 days).

Histologically, multiple scoring systems have been used in the past; the most frequently used systems are the WHO and Laurén classifications. This is the first study looking at the pathological characteristics of gastric carcinoma within this dog breed and its significance for the veterinary clinician. The histological evaluation of canine gastric carcinoma is frequently conducted with a scheme based on the 2010 WHO classification for human gastric carcinoma according to growth pattern [18]. Different growth patterns can exist within one tumour and the classification is based on the most dominant pattern present. Sometimes, both a papillary and tubular pattern are similarly present and in such cases the term tubulopapillary is used. Tubular carcinomas (1) can best be recognised by their duct-like branches, which can vary in amount of dilatation due to the volume of mucus build-up within the lumen, causing flattening of the cell morphology. This type of tumour carries a better prognosis than the other subtypes in humans. Papillary adenocarcinomas (2) can be clearly recognised by their finger-like projections (papillae) which are lined with cancerous cells, specifically cylindrical or cuboidal cells. The degree of differentiation can vary as well as the complexity of the branching structures. The third category consists of mucinous adenocarcinomas (3) which are defined by extracellular mucin pools made by glands on the epithelium of the tumour cells. These pools can result in visible gaps between the cells histologically. Over half of the mucinous carcinoma usually consists of mucin pools. These tumours are often poorly differentiated and have a worse prognosis than the other subtypes in humans. Other cancerous cells, such as signet-ring cells, can be found in these tumours as well. Signet-ring cell carcinomas (4) are carcinomas that are again defined by large masses of mucin; however, in this case, it is present within the cell as mucin vacuoles that push the nucleus to the periphery of the cell. If more than half of the tumour has cells with these characteristic intracytoplasmic mucin vacuoles, the tumour will be staged as a signet-ring cell carcinoma. In human patients, it has been proven that correct staging of the gastric tumour has great value for further prognosis and preferred treatment [17,18,21,34]. In contrast, using the WHO classification system for gastric carcinoma, there was no association between either age of onset or survival in this study, with the different subtypes analysed individually. Due to the multiple subgroups in this staging system, the number of dogs per group was insufficient to enable statistical comparisons. However, when comparing the most represented WHO classification (tubular) to the other WHO types grouped together, a significantly longer survival time was observed for the tubular type. Larger studies inclusive of more breeds are needed to demonstrate whether the WHO system has clinical relevance and applicability for gastric carcinoma in the Tervueren and Groenendael variants.

When considering the Laurén classification, the diffuse type of gastric cancer carcinoma carries a worse prognosis in humans [35,36] and a has distinct developmental pathway distinguishing it from the intestinal type [37,38]. The intestinal type is defined by glandular lumina or more rarely by tracts in the epithelia and substantial and varying cells that are often hyperchromatic. They often form well-defined masses. The cells are usually more neatly organised and cohesive than those in the diffuse type, with an unbroken formation between the cells. In the diffuse type, differentiation and arrangement of the cells is poor. Instead of a well-defined mass, these cells are usually arranged in small clusters. Either the clusters or individual cells then cause infiltration of the rest of the tissues. In humans, the diffuse type carries a significantly worse prognosis than the intestinal type [17,39]. In the current study, tumour type, based on the Laurén classification, was significantly associated with survival. Scoring canine GC according to the Laurén classification has comparable prognostic results to humans, with the diffuse type having a worse prognosis. Whether to use both Laurén and WHO classification methods on gastric GC slides remains a point of discussion. Usually, diffuse (Laurén) patterns of tumours were non-tubular (WHO) too, indicating a similar prognostic result. Further research in a larger cohort of dogs with GC is needed to study whether to prefer the WHO or the Laurén classification, with more cases within the different non-tubular types of the WHO classification.

Besides tumour classification, investigation of the inflammation status of the patient is thought to be of influence for prognostication in humans [22]. In humans, CD8+ T cells are associated with a better prognosis, whereas other cells, including tumour-associated macrophages and neutrophils, are associated with a poorer prognosis [40,41]. For this study, biopsies were scored as mildly, moderately, or severely inflamed based on inflammatory cell counts in the mucosa of the biopsies. However, no significance with either age of onset or survival was found. Further research regarding the inflammatory microenvironment in canine GC is required to better understand its role in the aetiology of GC, possibly providing new means of therapeutic management.

The involvement of the cardia and curvature minor were not associated with age of onset or survival time in this study. However, the number of cases with a gastric carcinoma involving the cardia was too low to make definitive statements as to whether they are a different clinicopathological subgroup. More research is needed to further examine the involvement of different locations within the stomach, including the fundus, curvature major, gastric body and antrum.

Mean age of onset was in concordance with previously reported case series of gastric carcinoma in Belgian shepherds [4] and other breeds [21,33]. In contrast to these earlier, smaller studies, we found that the sex distribution consisted of 29 males to 32 females; hence, there was no male predisposition to gastric carcinoma in Belgian Shepherds as has been previously reported [9].

The poor prognosis for gastric carcinoma in dogs is due to the aggressiveness and invasiveness of the condition, the very poor treatment options when diagnosed late in the stage of the disease and therefore also due to the limited and non-specific clinical signs [9]. Many dogs were presented for endoscopy or exploratory laparotomy at an advanced stage of the disease, with euthanasia frequently elected at or shortly after diagnosis. Twenty-one of the dogs were euthanised within two days of diagnosis, because of the poor prognosis in combination with severe clinical signs that compromised the welfare of the animals. These dogs still contributed to the analysis and, therefore, in this study, survival was defined as time from first clinical signs to death; however, the variation in detection of clinical signs and reporting specificity makes this parameter less reliable.

Unlike in the development of intestinal-type gastric carcinoma in humans, the role of helicobacter infection in dogs is questionable. Helicobacter has been reported with widely varying frequencies in canine gastric samples with and without gastric pathology [26,27,28,42]. In the present study, helicobacter was only investigated using routine H&E slides for the presence of typical helical bacteria. More sensitive and specific techniques for investigating the presence of helicobacter in gastric biopsies include PCR, immunohistochemistry and fluorescence in situ hybridization (FISH), and ancillary stains such as Giemsa would be desirable. Using H&E only may have led to underreporting; however, H&E can be sufficient for detecting routine helicobacter presence [43,44]. With only two gastric biopsies positive for helicobacter, of which only one was intraglandular, the involvement of helicobacter in the etiopathogenesis of gastric cancer in Belgian Shepherds is unlikely.

Limitations of the current study are due to sample size, especially with the small numbers of WHO-classified types other than the tubular type. A larger sample size of Belgian Shepherds would permit the confirmation of the histopathological scoring and additional breeds would be useful to determine whether the findings in the Belgian Shepherd were applicable to other breeds. Even with the absence of enough different subtypes, though, the better prognosis in tubular-classified GC is clear. In addition, the inclusion of more necropsy data would give more insight into the reproducibility and representative nature of gastric endoscopic biopsies.

The relatively high prevalence and poor prognosis within this breed suggests that exploring a genetic basis for the condition can provide insight into reducing the disease. To successfully conduct a genetic study, a clear phenotype must be further explored with the addition of relevant external factors for canine gastric carcinoma.

## 5. Conclusions

In this study consisting of 61 well-defined cases of gastric carcinoma in the Tervueren and Groenendael variants of the Belgian Shepherd dog breed, histological evaluation according to the Laurén classification has shown potential as a prognostic clinical tool for veterinarians and may prove to be important for genetic studies that require clear phenotype classification. Tervueren dogs had a younger age of onset compared to the Groenendael dogs; however, these breed variants were not associated with the different classification methods. The mean survival time for the diffuse type was 4 months shorter than for the intestinal type in the Lauren classification. In the WHO classification, non-tubular tumours showed a 4-month shorter mean survival time. Both observations are in line with prognoses based on these classifications in humans.

## Figures and Tables

**Figure 1 animals-13-01532-f001:**
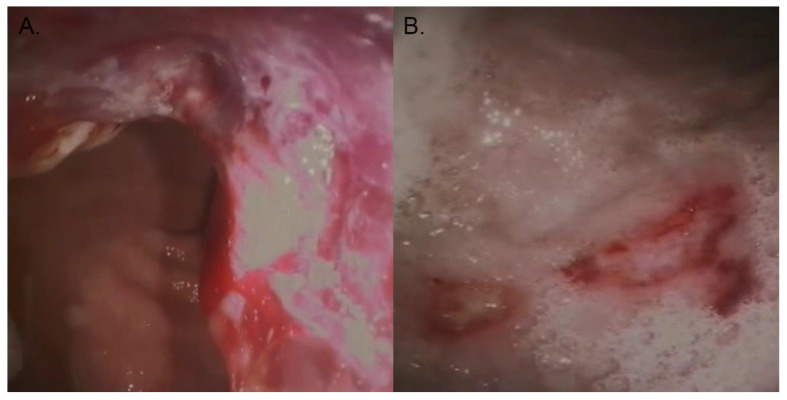
Photo courtesy of the authors. (**A**) Diagnostic imaging through endoscopy depicting canine gastric carcinoma. (**B**) Ulcerations on the gastric wall of an affected dog.

**Figure 2 animals-13-01532-f002:**
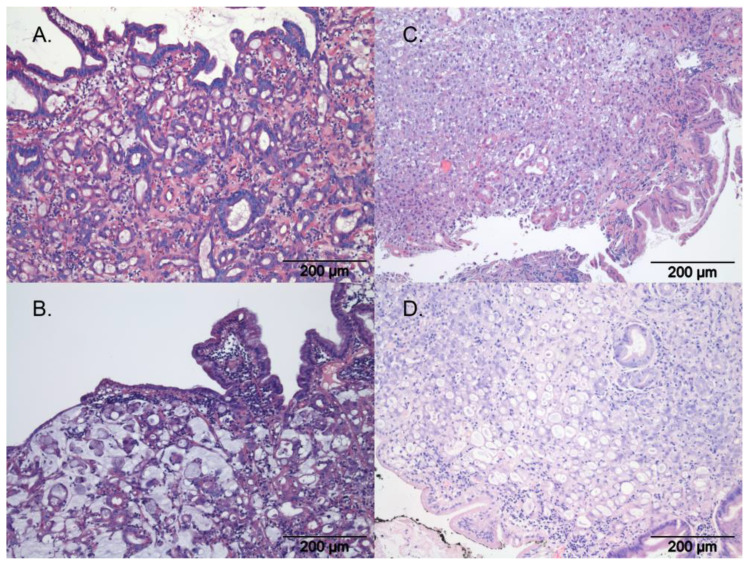
Histopathology of gastric carcinoma according to WHO, with H&E staining; photo courtesy of the authors. (**A**) Tubular gastric carcinoma, (**B**) mucinous gastric carcinoma, (**C**) undifferentiated gastric carcinoma, (**D**) signet-ring gastric carcinoma.

**Figure 3 animals-13-01532-f003:**
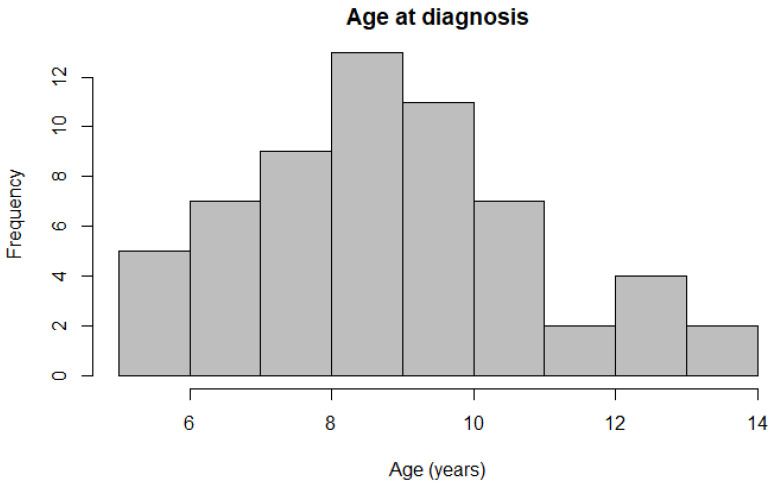
Histogram depicting the distribution of ages at diagnosis across all participating Belgian Shepherd dogs.

**Figure 4 animals-13-01532-f004:**
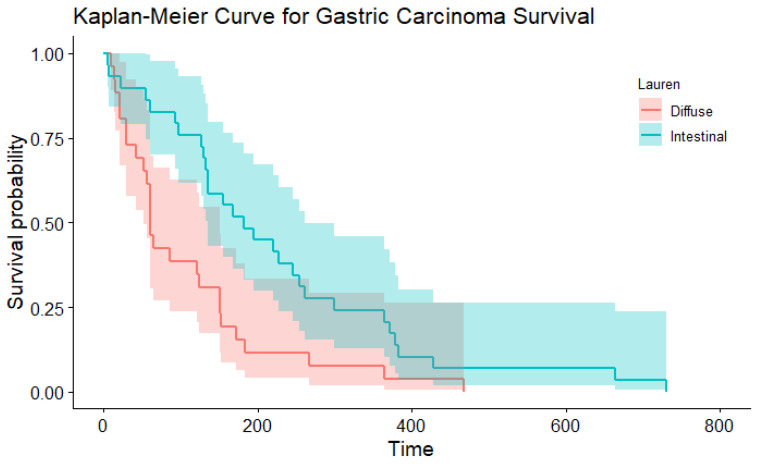
The Kaplan–Meier curve depicting survival based on the Laurén classification.

**Figure 5 animals-13-01532-f005:**
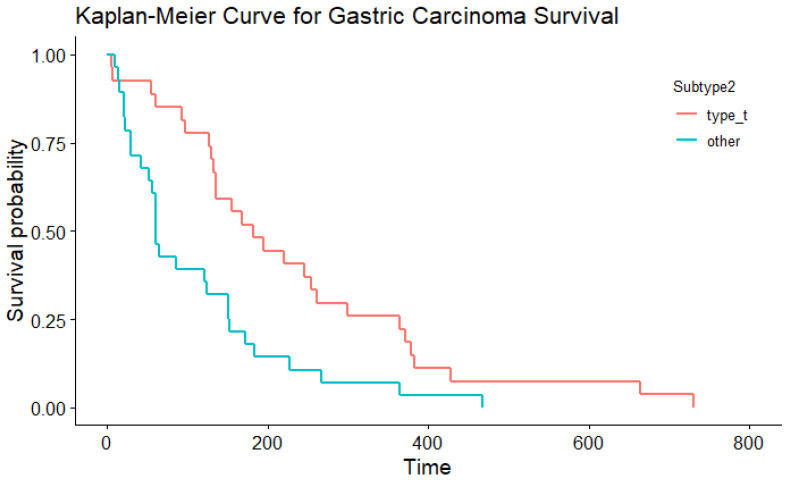
The Kaplan–Meier curve depicting survival based on the WHO classification.

**Table 1 animals-13-01532-t001:** Histological characteristics according to Head, K.W. et al., published by the Armed Forces Institute of Pathology in cooperation with the American Registry of Pathology and the World Health Organization Collaborating Center for Worldwide Reference on Comparative Oncology in 2003 [18]. Epithelial malignancies of the stomach, when recognised as adenocarcinoma, are first subdivided into tubular, papillary, tubulopapillary or signet-ring cell tumours. For this study, we also added the mucinous type. They are then classified as squamous cell carcinoma (not considered in this work) or undifferentiated carcinoma.

Tumour Type	Definition
Tubular	A malignant tumour composed of tubules embedded in connective tissue.
Papillary	A tumour in which the malignant epithelium is thrown up into papillary projections.
Tubulopapillary	A malignant tumour showing both tubular and papillary features.
Mucinous	Intracytoplasmic acid mucin-containing vacuoles; goblet cell appearance (i.e., single large vacuole); eosinophilic granules of neutral mucin
Signet-ring	A malignant tumour in which more than half is composed of epithelial cells containing intracellular acid or neutral mucin, displacing the nucleus to one side of the cells. Poor differentiation
Undifferentiated	A malignant tumour that has epithelial characteristics with no evidence of squamous or glandular differentiation.

**Table 2 animals-13-01532-t002:** Number of samples and available data per variable. Abbreviations: C—cardia; CM—curvature minor; M—male; MN—male neutered; F—female; FN—female neutered.

	Sample Count with Tumour Location	Sex	Age of Onset (Diagnosis)	*p*-Value with Age of Onset (Diagnosis)
Tervueren	45 (2 C, 34 CM)	10 M, 11 MN, 11 F, 12 FN	8.5 ± 1.90	*p* = 0.009
Groenendael	16 (2 C, 9 CM)	3M, 11MN, 1MN-unknown, 1 F, 8FN	10.1 ± 2.01	
Total	61		60	

**Table 3 animals-13-01532-t003:** Results of the univariable regression analyses.

Dependent Variable (Age of Onset)	Univariable Regression	
Category	Estimate	*p*-Value
Groenendael Breed	1.5312	0.009
Tumour (location CM yes)	0.6627	0.396
Tumour (location cardia yes)	−0.3185	0.777
Sex (female)	−0.06229	0.907
Neuter status (neutered)	−0.1631	0.767
WHO subtype (subtype other)	0.07119	0.894
Laurén classification (intestinal)	−0.1958	0.714
Inflammation (mild)	0.7660	0.478

**Table 4 animals-13-01532-t004:** WHO and Laurén classifications strongly associate with one another (*p* = 2793 × 10^−12^), with all intestinal (Laurén)-type tumours being either tubular or tubulopapillary according to the WHO scheme. Abbreviations: I—intestinal; D—diffuse.

	WHO and (Laurén)
Mucinous	12 (D = 12, I = 0 )
Signet-ring	5 (D = 5, I = 0)
Tubular	29 (D = 0, I = 29)
Papillary	0
Tubulopapillary (mixed)	2 (D = 0, I = 2)
Unclassified	12 (D = 12, I = 0)

**Table 5 animals-13-01532-t005:** Survival over time according to the Laurén classification, with 26 dogs in the diffuse category and 29 dogs in the intestinal category.

Survival Time Laurén = Diffuse						
Time in Days	n Survived	n Casualties	Survival	Std.Err	Lower 95% CI	Upper 95% CI
0	26	0	1.000	0.000	1.000	1.000
100	10	16	0.385	0.0954	0.237	0.625
200	3	7	0.115	0.0627	0.0398	0.334
300	2	1	0.077	0.0523	0.020	0.291
400	1	1	0.039	0.0377	0.006	0.263
**Survival time Laurén = intestinal**						
**Time in days**	**n survived**	**n casualties**	**Survival**	**Std.err**	**Lower 95% CI**	**Upper 95% CI**
0	29	0	1.000	0.000	1.000	1.000
100	22	7	0.756	0.080	0.618	0.932
200	13	9	0.448	0.0923	0.299	0.671
300	8	6	0.241	0.080	0.127	0.460
400	3	4	0.103	0.057	0.035	0.302
500	2	1	0.069	0.047	0.018	0.263
600	2	0	0.069	0.047	0.018	0.263
700	1	1	0.035	0.034	0.005	0.237

**Table 6 animals-13-01532-t006:** Survival over time according to the WHO classification, consisting of 27 dogs in the tubular category and 28 dogs in a primarily non-tubular category (subtype 2).

Survival Time WHO = Tubular						
Time in Days	n Survived	n Casualties	Survival	STD.Err	Lower 95% CI	Upper 95% CI
0	27	0	1.000	0.000	1.000	1.000
100	21	6	0.778	0.080	0.636	0.952
200	12	9	0.444	0.096	0.292	0.678
300	8	5	0.259	0.084	0.137	0.490
400	3	4	0.111	0.061	0.038	0.323
500	2	1	0.074	0.050	0.020	0.281
600	2	0	0.074	0.050	0.020	0.281
700	1	1	0.037	0.036	0.005	0.253
**Survival time WHO = primarily non-tubular**						
**Time in days**	**n survived**	**n casualties**	**Survival**	**Std.err**	**Lower 95% CI**	**Upper 95% CI**
0	28	0	1.000	0.000	1.000	1.000
100	11	17	0.393	0.092	0.248	0.623
200	4	7	0.143	0.066	0.058	0.354
300	2	2	0.071	0.049	0.019	0.272
400	1	1	0.036	0.035	0.005	0.245

## Data Availability

The datasets used can be acquired through the corresponding author.

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
