# Peer review of "The Histopathological Characteristic of Gastric Carcinoma in the Belgian Tervueren and Groenendael Dog: A Comparison of Two Classification Methods"

_animals, 2023, doi:10.3390/ani13091532_

Round 1

Reviewer 1 Report

This paper discusses  histopathological Characteristics of Gastric Carcinoma in the Belgian Tervueren and Groenendael Dog.  Clinical parameters and survival time  after diagnosis was recorded and was investigated in relation to tumor classification.

The study is interesting but requires careful revision.

In the materials and methods it is not clear how many are biopsy samples and how many are necropsy samples.

I think it is important that quality and adequacy is assessed for biopsy samples.

J.A. Cartwright, T.L. Hill, S. Smith, and D. Shaw Evaluating Quality and Adequacy of Gastrointestinal Samples Collected using Reusable or Disposable Forceps. J Vet Intern Med 2016;30:1002–1007.

Two different classifications are considered in the study Laurén (1965) and WHO (2003)

For the second in the text (table 3) reference is made as a bibliography to Histological characteristics according to Head K.W., Else R.W., Dubbielzig R.R. in Tumours 218 in domestic animals. 4th Edition (2003). Eds Meuten D.G. pp 454, which I did not find in the list of bibliography, where Head, K.W. is cited instead. (the other authors, Cullen, Dubilezig, Else, Misdorp, Patnaik Tateyama and van der Gaag are missing) Histological Classification of Tumors of the Alimentary System of Domestic Animals; Published by the Armed Forces Institute of Pathology in cooperation with the American Registry of Pathology and the World Health Organization: 2003.

I think it is more correct to use the latter bibliographic entry, which is the most widely used in histopathological diagnostics.

In this case, table 3 must be rewritten on the basis of the definitions in Head, Cullen et al. 2023)

Epithelial malignancies are subdivided into:

Adenocarcinoma

A malignant glandular tumour with no distinguishing features that would place it in a more specific category

AC is subdivided into:

-Tubular

A malignant tumour composed of tubules embedded in connective tissue.

-Papillary

A tumour in which the malignant epithelium is thrown up into papillary projections.

-Tubulopapillary

A malignant tumour showing both tubuar and papillary features.

-Signet-ring cell

A malignant tumour in which more than half is composed of epithelial cells containing intracellualr acid or neutral mucin, displacing the nucleus to one side of the cells.

They are then classified

-Squamos cell carcinoma (not considered in the work)

A malignant tumour composed of cells with features of squamous epithelium

and

Undifferentiated carcinoma

A malignant tumour that has epithelial characteristics with no evidence of squamous or glandular differentiation.

The images included in the study for the various tumour types are not of good quality and should be taken at a higher magnification.

The fact of using two different types of classification, in my opinion, makes the study confusing in terms of interpretation. My advice is to use the most recent WHO classification scheme of 2003, citing Lauren (1965) but then not using it in the statistical evaluation.

In the text it is written that all slides of gastric carcinoma biopsies were scored according to the WHO classification according to the criteria in Table 1. But Table 1 refers to Reference values for inflammatory cell counts per 250 mm length of mucosa, through a 135 high-power field microscope (HPF) in both the gastric body and gastric antrum of dogs with gastric 136 carcinoma, as published by Day et al. 2008.

In reference to the appearance of the inflammatory component it is correct to use Day's work but I think it is more useful to refer to the score:

-Normal intraepithelial lymphocytes

Sparse population of approximately 1-2 cells per stretch of 50 epithelial cells.

-Mild increase in intraepithelial lymphocytes

Individual lymphocytes, up to 10 per stretch of 50 epithelial cells.

-Moderate increase in intraepithelial lymphocytes

Lymphocytes may cluster in groups of up to 4 cells. There may be up to 20 per stretch of 50 epithelial cells.

-Marked increase in intraepithelial lymphocytes

Epithelium is more diffusely infiltrated by lymphocytes (up to 50 per stretch of 50 epithelial cells)

The same applies to the neutrophilic and eosinpholic granulocyte component.

To summarise, I am of the opinion, an improvement in the study could be had by adding an assessment of the quality and accuracy of the biopsy specimens, indicating how many specimens are biopsy and how many are necropsy or intraoperative. Use the 2003 WHO classification, citing the bibliographic entry correctly. Do not use Lauren's 1965 classification, which in my opinion only creates confusion and makes the results difficult to interpret.

Author Response

Dear Reviewer,

Through this Word file, I would like to address the changes made to the manuscript. The manuscript has been changed to better display our main aim of study: to compare two classification systems for gastric carcinoma in Belgian Shepherds, namely WHO and Laurén. Additionally, information has been added to provide more insight into some choices within this study, like the locations within the stomach, revision of spelling mistakes has been done too. With the following text I would like to address the feedback given by the reviewers.

Reviewer 1 

This paper discusses histopathological Characteristics of Gastric Carcinoma in the Belgian Tervueren and Groenendael Dog.  Clinical parameters and survival time  after diagnosis was recorded and was investigated in relation to tumor classification.

The study is interesting but requires careful revision.

In the materials and methods it is not clear how many are biopsy samples and how many are necropsy samples.

This information will be added. Samples acquired by biopsy and samples acquired by necropsy (large samples) were documented.

I think it is important that quality and adequacy is assessed for biopsy samples.

J.A. Cartwright, T.L. Hill, S. Smith, and D. Shaw Evaluating Quality and Adequacy of Gastrointestinal Samples Collected using Reusable or Disposable Forceps. J Vet Intern Med 2016;30:1002–1007.

Scoring for gastric slide adequacy was not mentioned in the paper as there were no inadequate samples present or used in the study. When a slide turned out to be low in quality, new slides were obtained from the biopsies.

Two different classifications are considered in the study Laurén (1965) and WHO (2003)

For the second in the text (table 3) reference is made as a bibliography to Histological characteristics according to Head K.W., Else R.W., Dubbielzig R.R. in Tumours 218 in domestic animals. 4th Edition (2003). Eds Meuten D.G. pp 454, which I did not find in the list of bibliography, where Head, K.W. is cited instead. (the other authors, Cullen, Dubilezig, Else, Misdorp, Patnaik Tateyama and van der Gaag are missing) Histological Classification of Tumors of the Alimentary System of Domestic Animals; Published by the Armed Forces Institute of Pathology in cooperation with the American Registry of Pathology and the World Health Organization: 2003.

I think it is more correct to use the latter bibliographic entry, which is the most widely used in histopathological diagnostics.

The reference will be edited to follow our bibliography.

In this case, table 3 must be rewritten on the basis of the definitions in Head, Cullen et al. 2023)

Epithelial malignancies are subdivided into:

Adenocarcinoma

A malignant glandular tumour with no distinguishing features that would place it in a more specific category

AC is subdivided into:

-Tubular

A malignant tumour composed of tubules embedded in connective tissue.

-Papillary

A tumour in which the malignant epithelium is thrown up into papillary projections.

-Tubulopapillary

A malignant tumour showing both tubular and papillary features.

-Signet-ring cell

A malignant tumour in which more than half is composed of epithelial cells containing intracellualr acid or neutral mucin, displacing the nucleus to one side of the cells.

They are then classified

-Squamos cell carcinoma (not considered in the work)

A malignant tumour composed of cells with features of squamous epithelium

and

Undifferentiated carcinoma

A malignant tumour that has epithelial characteristics with no evidence of squamous or glandular differentiation.

The table has been rewritten according to the advice above. However, we have also considered mucinous subtypes of adenocarcinoma. As we have many (12) cases with this type, we find it important to mention it in the table as well. We are a bit puzzeled with your reference to Head, Cullen et al. 2023 as we could not find this. Head 2003 could be found.

The images included in the study for the various tumour types are not of good quality and should be taken at a higher magnification.

We understand that images taken at a higher magnification are preferred. We were not able to retake these in time, but will be able to provide them on a later occasion. We ask for your understanding, thank you.

The fact of using two different types of classification, in my opinion, makes the study confusing in terms of interpretation. My advice is to use the most recent WHO classification scheme of 2003, citing Lauren (1965) but then not using it in the statistical evaluation.

As no gold standard for histologically evaluating canine gastric carcinoma in Belgian Shepherd dogs has been described as of yet, we find it important to include both classification methods in full detail. While we understand that the most recent WHO classification might be of preference in practice, including both gives both the veterinary clinician and the veterinary pathologist the opportunity to make an informed choice on how to classify these tumors. Not further using our data regarding the Laurén classification might arise confusion and questions, for example, if Laurén is unusable or undesired as a method. As we did have a significant difference in onset of symptoms using Laurén, as well as when using WHO, we find it important to include both.

In the text it is written that all slides of gastric carcinoma biopsies were scored according to the WHO classification according to the criteria in Table 1. But Table 1 refers to Reference values for inflammatory cell counts per 250 mm length of mucosa, through a 135 high-power field microscope (HPF) in both the gastric body and gastric antrum of dogs with gastric 136 carcinoma, as published by Day et al. 2008.

This is true, the reference was mistakenly made to table 1; it should have been table 3. Table 3 has now changed locations and became table 1.

In reference to the appearance of the inflammatory component it is correct to use Day's work but I think it is more useful to refer to the score:

-Normal intraepithelial lymphocytes

Sparse population of approximately 1-2 cells per stretch of 50 epithelial cells.

-Mild increase in intraepithelial lymphocytes

Individual lymphocytes, up to 10 per stretch of 50 epithelial cells.

-Moderate increase in intraepithelial lymphocytes

Lymphocytes may cluster in groups of up to 4 cells. There may be up to 20 per stretch of 50 epithelial cells.

-Marked increase in intraepithelial lymphocytes

Epithelium is more diffusely infiltrated by lymphocytes (up to 50 per stretch of 50 epithelial cells)

The same applies to the neutrophilic and eosinpholic granulocyte component.

We made the table with inflammatory cell counts based on Day et al. 2008 with the aim of giving an overview in the differences in normal values, also taking into account the location of the stomach. However, no exact cell counts were done for this study, and scoring was more based on the subjective opinion of the pathologist, while taking into account Day et al. As no exact cell counts were made, we feel it is more appropriate for us to delete this table. We will add the advised scoring explanation to the manuscript, as we feel it helps with the understanding of how we performed the inflammation scoring, based on Day’s work better than the cell count table.

To summarise, I am of the opinion, an improvement in the study could be had by adding an assessment of the quality and accuracy of the biopsy specimens, indicating how many specimens are biopsy and how many are necropsy or intraoperative. Use the 2003 WHO classification, citing the bibliographic entry correctly. Do not use Lauren's 1965 classification, which in my opinion only creates confusion and makes the results difficult to interpret.

We appreciate the in-depth feedback and have added many changes that we feel enhance the paper. However, we strongly feel that including the Laurén classification is important to this paper as it gives the clinician and pathologist information regarding this method with canine GC that has not been published before.

With sincere thanks to you.

Christina Kijan & Paul Mandigers

Reviewer 2 Report

Congratulations on an excellent retrospective study, presenting a very good introduction, a clear description of the methods, interesting results, and excellent discussion. It contributes to the awareness of the clinical practician for differences in the age of onset of gastric carcinoma in Tervuren and Groenendael Belgian Shepherd dogs, and aids in establishing prognosis, but it is equally interesting to more specialized readers.

Author Response

Dear reviewer,

Thank you for your review and comments

Reviewer 3 Report

The title of the manuscript is "The Histopathological Characteristics of Gastric Carcinoma in 2 the Belgian Tervueren and Groenendael Dog" but the manuscript also includes clinical parameters and survival time after diagnosis in relation to tumor classification (which means that the title does not fully convey the content of the manuscript). Because of this, there is an impression that it is a retrospective study that wants to present the results in the form of a prospective study. For this reason, the authors should clearly show in the material and methods the origin of the samples (exactly from which institutions they come) and how the data related to the clinical picture, symptoms and time of the animal's death were collected (in lines 184-185 it is stated that these data were obtained from the owner, but it is not defined how - by questionnaire, by phone, during a regular visit to the clinic or in some other way). Perhaps the paper would be more relevant if it would be simply stated that it is a retrospective analysis of gastric tumors in Belgian Shepherd dogs based on clinical and pathological data collected between 2003 and 2017. 

Furthermore, it is not clear what the authors wanted to show with the statistical data related to he significance of the veterinarians involved in the biopsies (lines 140-143 and 163-165).

Also, Figures 1 and 2 probably represent the macroscopic and histological appearance of the lesion in the dogs that are part of the study. Therefore, these figures should be presented in the chapter "Results" (and not in the  chapter "Introduction"). Furthermore, in Figure 2, Figure 2B appears to be a mucinous carcinoma and Figure 2C appears to be an undifferentiated carcinoma.

Line 40 - it is suggested to add the word "carcinoma" to the keywords.

Author Response

Dear Reviewer,

Through this Word file, I would like to address the changes made to the manuscript. The manuscript has been changed to better display our main aim of study: to compare two classification systems for gastric carcinoma in Belgian Shepherds, namely WHO and Laurén. Additionally, information has been added to provide more insight into some choices within this study, like the locations within the stomach, revision of spelling mistakes has been done too. With the following text I would like to address the feedback given by the reviewers.

Reviewer 3
The title of the manuscript is "The Histopathological Characteristics of Gastric Carcinoma in 2 the Belgian Tervueren and Groenendael Dog" but the manuscript also includes clinical parameters and survival time after diagnosis in relation to tumor classification (which means that the title does not fully convey the content of the manuscript).

Most of the clinical parameters are relevant for our follow-up study. What is relevant now, is the onset of both the symptoms (as described by the owner) and the diagnosis. The notion of clinical parameters will be removed to not give false expectations.

Because of this, there is an impression that it is a retrospective study that wants to present the results in the form of a prospective study. For this reason, the authors should clearly show in the material and methods the origin of the samples (exactly from which institutions they come) and how the data related to the clinical picture, symptoms and time of the animal's death were collected (in lines 184-185 it is stated that these data were obtained from the owner, but it is not defined how - by questionnaire, by phone, during a regular visit to the clinic or in some other way).

Data regarding the dogs and their symptoms were obtained by questionnaire, data regarding the tumor and diagnosis (when external) were obtained by a question form or patient data from the practitioner. This will be added to the manuscript. However, it is not relevant or practical to show all the institutions external samples came from. During statistical analysis, the involvement of different veterinarians did not turn out as significant, also, most referral cases came from different clinics. The exact number will be added (13 referral veterinarians).

Perhaps the paper would be more relevant if it would be simply stated that it is a retrospective analysis of gastric tumors in Belgian Shepherd dogs based on clinical and pathological data collected between 2003 and 2017. 

The title will be changed to be more relevant to the study: which is a retrospective analysis comparing two histopathological classification methods for gastric carcinoma in Belgian Shepherd dogs.

Furthermore, it is not clear what the authors wanted to show with the statistical data related to he significance of the veterinarians involved in the biopsies (lines 140-143 and 163-165).
The potential issue was that results / outcomes etc could have been affected by the vet that saw the case.  So, whilst we weren’t interested in the exact effect itself (from a scientific point of view), we needed to take this into account in the models.  So, we added this as a random effect.

Also, Figures 1 and 2 probably represent the macroscopic and histological appearance of the lesion in the dogs that are part of the study. Therefore, these figures should be presented in the chapter "Results" (and not in the  chapter "Introduction").
While yes, the pictures are made during this study and from participating patients, having these in the introduction helps the reader visualize the problem early on and provides an overview of the histological differences between the WHO subtypes as this is a major part of the paper, and named in the introduction. Especially the endoscopic images can be relevant in the introduction, as we don’t delve deep into the macroscopy of GC in the rest of the paper.

Furthermore, in Figure 2, Figure 2B appears to be a mucinous carcinoma and Figure 2C appears to be an undifferentiated carcinoma.
This will be corrected in the manuscript.

Line 40 - it is suggested to add the word "carcinoma" to the keywords.
This will be added.

Thank you for your feedback.

Christina Kijan & Paul Mandigers

Round 2

Reviewer 1 Report

The authors improved their paper.

I only have a few additional comments

Lauren's classification was applied to gastric carcinomas in humans.

Considering that Lauren's classification is important for the article, I think it is useful to point out that in 1976 based on Lauren's classification  a histologic classification of lower alimentary tract tumors in different species of domestic animals, including the dog, has been developed by the World Health Organization (WHO).

In 1978 Patnaik et al made a similar comparative study and histologic classification of 26 canine gastric adenocarcinomas. We found th at the two main histologic types of human gastric adenocarcinoma occurred also in dogs.

I think it is useful to include these two bibliographic entries, confirming that Laurén classification had been applied to gastric carcinoma in dogs.

HEAD, K.W .: Tumors of the lower alimentary tract. Bull WHO 53:1 67-1 86.1 976

AK. Patnaik. A. I. Hurvitz/. and G. F. Johnson Vet. Pathol. 15: 600-607 ( I978)

Regarding the pictures, I am not sure if the 200 microns bar is representative of the magnification used. Please check.

In the legend, please enter what type of stain was used

Author Response

Dear Reviewer,

First, we thank you very much for taking your time to revise our article regarding gastric carcinoma in Belgian Shepherd dogs a second time. We appreciate that the reviewers stated that the changes made during the major revisions improved the work. Through this Word file, I would like to address the most recent changes made to the manuscript, during the minor revisions.

Reviewer 1: The authors improved their paper.

I only have a few additional comments

Lauren's classification was applied to gastric carcinomas in humans.

Considering that Lauren's classification is important for the article, I think it is useful to point out that in 1976 based on Lauren's classification  a histologic classification of lower alimentary tract tumors in different species of domestic animals, including the dog, has been developed by the World Health Organization (WHO). In 1978 Patnaik et al made a similar comparative study and histologic classification of 26 canine gastric adenocarcinomas. We found th at the two main histologic types of human gastric adenocarcinoma occurred also in dogs.

I think it is useful to include these two bibliographic entries, confirming that Laurén classification had been applied to gastric carcinoma in dogs.

HEAD, K.W .: Tumors of the lower alimentary tract. Bull WHO 53:1 67-1 86.1 976

  1. Patnaik. A. I. Hurvitz/. and G. F. Johnson Vet. Pathol. 15: 600-607 ( I978)

Thank you for this useful feedback and these articles. After going over them, we agree that these are important to refer to in our manuscript. We have added this information to our manuscript, as well as the references to our bibliography in the appropriate format, while adjusting the rest of the bibliography accordingly.

Regarding the pictures, I am not sure if the 200 microns bar is representative of the magnification used. Please check.In the legend, please enter what type of stain was used.

The used stain was H&E staining, this information has been added to the legend. The 200 microns bar is indeed representative.

Reviewer 3 Report

Thanks to the authors for taking into account the above suggestions for improving the manuscript. The aims of the study are more clearly presented. 

Figures 1 and 2 are still left in the introduction chapter and the source of those pictures is not indicated. If the figures are the property of authors and were created during the diagnostic procedure, I think they should be shown with the results (gastroscopy (2.2) and histopathology (2.4)).

Author Response

Dear reviewer, thank you for your review.

Reviewer 3: Thanks to the authors for taking into account the above suggestions for improving the manuscript. The aims of the study are more clearly presented. 
Figures 1 and 2 are still left in the introduction chapter and the source of those pictures is not indicated. If the figures are the property of authors and were created during the diagnostic procedure, I think they should be shown with the results (gastroscopy (2.2) and histopathology (2.4)).

Thank you for the feedback. Both the endoscopic and the microscopic pictures are taken by the authors as part of the study. The gastroscopic images were indeed during the diagnostic procedure of a case used in this study. We understand the preference of having these elsewhere, as they weren’t acquired from other papers but resulted from this research. Their position in the manuscript will be changed accordingly, to illustrate the procedures described in subheadings 2.2 and 2.4.

Other than the changes above, minor changes have been made to correct some linguistic errors and to better explain the aim of the paper, specifically the addition of tumour location in the analysis. Also, some table-specific abbreviations were added to have a better understanding of said tables (table 2, table 3).

We hope these changes further improve the paper and will gladly answer any questions that might arise. We appreciate your time and efforts.